# Graph Learning with Distributional Edge Layouts

## Abstract

Graph Neural Networks (GNNs) learn from graph-structured data by passing local messages between neighboring nodes along edges on certain topological layouts. Typically, these topological layouts in modern GNNs are deterministically computed (e.g., attention-based GNNs) or locally sampled (e.g., GraphSage) under heuristic assumptions. In this paper, we for the first time pose that these layouts can be globally sampled out of steady-state graphs following Boltzmann distribution equipped with explicit physical energy, leading to more viable pairwise distance configurations in the physical world. We argue that a collection of sampled/optimized layouts can capture the wide energy distribution and better characterize the intrinsic properties of input topology, therefore easing downstream tasks. As such, we propose Distributional Edge Layouts (DELs) to serve as a complement to a variety of GNNs. DEL is a pre-processing strategy independent of subsequent GNN variants, thus being highly flexible. Experimental results demonstrate that DELs consistently and substantially improve a series of GNN baselines, achieving state-of-the-art performance on multiple datasets. DEL is open-sourced at `https://anonymous.4open.science/r/DEL`.

## 1 Introduction

Graph neural networks (GNNs), which learn informative knowledge from graph-structured data, have witnessed remarkable success in a large variety of applications, such as molecular generation (Jin et al., 2018), link prediction (Zhang & Chen, 2018), to name a few. A standard learning paradigm in GNNs is the message-passing mechanism, which takes node/edge features as inputs and propagates messages through neighboring nodes defined on some topological layouts (Wu et al., 2020). Given a graph $\mathcal{G} = (\mathbf{H}^{(0)}, \mathbf{A})$ where $\mathbf{H}^{(0)}$ and $\mathbf{A}$ are respectively initial features and connectivity, GNNs seek to learn meaningful representations by stacking multiple unit layers. Concretely, for almost all variants of GNNs, an update layer can be generalized as (Balcilar et al., 2021a):

$$\mathbf{H}^{(l+1)} = \sigma \left( \sum_s \mathbf{C}^{(s)} \mathbf{H}^{(l)} \mathbf{W}^{(l,s)} \right) \tag{1}$$

where $\mathbf{H}^{(l)} \in \mathbb{R}^{n \times f_l}$ refers to the feature at $l$-th layer. $\mathbf{C}^{(s)} \in \mathbb{R}^{n \times n}$ is the $s$-th topological layout on which messages are propagated between neighboring nodes, and $\mathbf{W}^{(l,s)} \in \mathbb{R}^{f_l \times f_{l+1}}$ is the learnable parameter at the $l$-th layer and $s$-th layout. $\sigma(\cdot)$ is a non-linear activation function. It is argued in Bouritsas et al. (2022) that GNNs generally differ from each other by the choice of layouts $\mathbf{C}^{(s)}$.

Early GNNs utilize a direct mapping of original input connectivity $\mathbf{A}$ as the topological layout (Kipf & Welling, 2016; Yu et al., 2019), which may result in sub-optimal performance. Stronger GNNs in general focus on designing a more sophisticated layout set $\mathbf{C}^{(s)}$ to decouple the computational layout from the raw graph, bringing about benefits for downstream tasks accordingly. Several aspects have been considered in this line of methods, such as spectral information (Huang et al., 2022), expressivity (Xu et al., 2019b), and attention mechanism (Veličković et al., 2018; Wu et al., 2021; Rampášek et al., 2022). Among these branches, attention-based GNNs may be the most successful, delivering state-of-the-art performance on a range of public datasets (Rampášek et al., 2022; Zhang et al., 2022). Subgraph sampling, in parallel, is also employed to produce topological layouts, by incorporating locality-based randomness, such as random walk (Hamilton et al., 2017) and edge rewiring/dropping (Rong et al., 2020).

We note that existing GNNs obtain topological supports either through deterministic mappings or by injecting randomness locally. In the real world, however, layouts associated with a given graph may span over a wide potential distribution under some global criteria. For example, 3D conformations given a molecule atom-bond graph tend to exist in the physical world following Boltzmann distribution with a global free energy (Xu et al., 2021). Thus, deriving a single conformation is insufficient to holistically capture the potential energy surface, bringing about difficulty for subsequent tasks. Motivated by this, in this paper, we propose to cope with layouts by introducing an associated distribution over $\mathbf{C}$:

$$\mathbf{H}^{(l+1)} = \sigma \left( \int \mathbf{C} \mathbf{H}^{(l)} \mathbf{W}^{(l)} \mathrm{d}\mathbb{P}(\mathbf{C}) \right) \tag{2}$$

where $\mathbb{P}(\mathbf{C})$ is the probability density of occurrence of $\mathbf{C}$ and is related to some global energy function $E(\mathbf{C})$. Now two problems remain: 1) what energy and distribution to use; and 2) how to integrate derived layouts into GNNs.

To this end, we draw inspiration from energy-based layout computation in graph visualization (Fruchterman & Reingold, 1991; Kamada et al., 1989; Gansner et al., 2005; Wang et al., 2017). In a nutshell, layouts in graph visualization are diverse and viable configurations in low-dimensional space, explicitly optimized over physics-driven energy functions. A nice property is that it only depends on pure topology/connectivity without requiring node features, which can be flexibly coupled with multiple GNNs. Further motivated by statistical mechanics, we employ Boltzmann distribution defined on such energy to characterize the distribution of layouts $\mathbf{C}$. To sample from Boltzmann distribution, one can employ Langevin Dynamics to inject proper noise along the optimization trajectory. Empirically, we found that as long as the initial state of layouts is sufficiently random, the noise-free optimization can lead to diverse layouts and deliver significant performance in the downstream tasks. Furthermore, such a collection of layouts can be pre-calculated before fed into standard GNNs. In summary, we propose Distributional Edge Layouts (DELs), which sample layouts from the designated distribution, and flexibly inject this information into a series of GNNs as supplements. Experimentally, we observe that DEL can improve the learning performance of selected GNNs by a large margin on a wide range of datasets, demonstrating strong applicability and flexibility. In conclusion, our contributions are:

- We propose a generic GNN format by sampling topological layouts from Boltzmann distribution with explicit energy surface, which can capture a wide spectrum of global graph information.
- Calculation of DELs is independent of GNNs, leading to high flexibility and applicability.
- As a plug-in component, DELs substantially improve selected GNNs, achieving state-of-the-art performance on a variety of datasets.

## 2 RELATED WORK

**Graph Neural Networks with topological layout.** It has been argued by Bouritsas et al. (2022); Balcilar et al. (2021b) that the topological layout, represented as $\mathbf{C}$, determines the specific way of message passing between nodes. By default, connectivity $\mathbf{A}$ itself defines a layout, while a naive layout yields unsatisfactory performance. A series of GNNs then seek to directly transform $\mathbf{A}$, from either spatial or spectral perspectives (Kipf & Welling, 2016; Xu et al., 2019a; Defferrard et al., 2016; Dwivedi & Bresson, 2021; Kreuzer et al., 2021). Sampling/rewiring-based GNNs, in parallel, are further proposed by advocating local randomness on $\mathbf{C}$, bringing about higher robustness (Alon & Yahav, 2020; Topping et al., 2022; Rong et al., 2020). Many other GNNs also consider deriving $\mathbf{C}$ from an "edge feature" aspect, as $\mathbf{C}$ inherently aligns with some edge layout. For instance, EGNN (Gong & Cheng, 2019) tackles this challenge by constructing edge features based on the direction of directed edges in citation networks. Similarly, Sun et al. (2022) proposes leveraging chemical properties like bond types and bond directions to create edge features for molecular graphs. Circuit-GNN, introduced by Zhang et al. (2019), is tailored for distributed circuit design. In Circuit-GNN, nodes represent resonators, and edges denote electromagnetic couplings between pairs of resonators. These edge features are derived from physical circuit characteristics, including gap, shift, and relative position between square resonators, among other factors, which contribute to the refinement of topological layouts. Although some general edge feature construction schemes exist, such as MPNN (Gilmer et al., 2017), Point-GNN (Shi & Rajkumar, 2020), and CIE (Yu et al.,

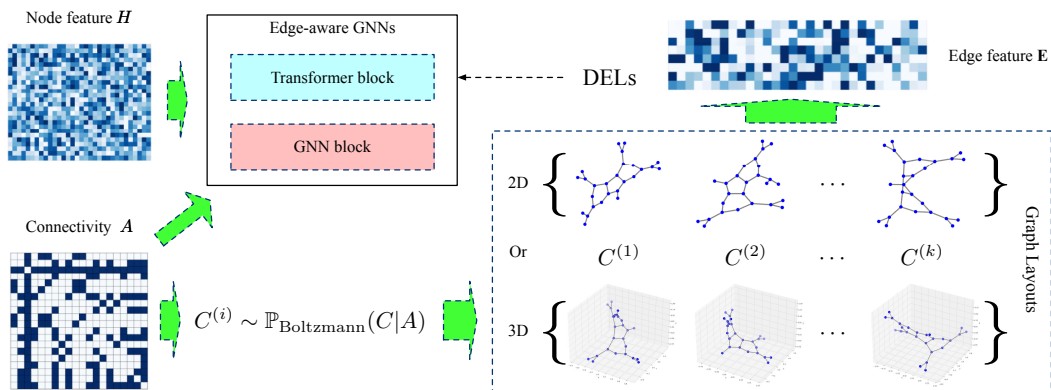

Figure 1: DEL pipeline. Our framework can be summarized into the following three steps: 1) **Sampling Layouts**: Sampling a set of layouts with local minimum energy for a given connectivity based on the Boltzmann distribution with the help of the energy-based layout algorithms. 2) **Constructing Edge Features**: Once we have a set of sampled layouts, we construct edge features based on these layouts. These edge features are designed to capture the potential energy surface of the system. Models can gain insights into the potential energy landscape and identify regions of high or low energy through the edge features. 3) **Edge-Aware GNNs**: In this final step, we combine the constructed edge features with a GNN backbone and use edge-aware GNNs to improve the performance of downstream tasks.

2019), they generate edge features through node features in a heuristic fashion without physical implication.

**Graph layout algorithm.** Graph layout algorithms arrange nodes and edges in low-dimensional space to depict viable graph topology. It is widely applicable in domains like social network analysis (Henry et al., 2007; Wang et al., 2020) and bioinformatics (Sych et al., 2019). Our primary focus is on energy-based layout algorithms (Fruchterman & Reingold, 1991; Kamada et al., 1989; Gansner et al., 2005; Wang et al., 2017), which adapt well to different topologies by simulating the physical interactions between nodes to determine their positions. These methods, rooted in principles of physics, offer a range of configurations in lower-dimensional spaces. Classic energy-based graph layout algorithms typically utilize either a spring model (Kamada et al., 1989; Gansner et al., 2005) or a spring-electrical model (Fruchterman & Reingold, 1991; Jacomy et al., 2014). In the spring model, edges are treated as elastic connections, and relationships between nodes are simulated by the tension in the springs. The spring-electrical model combines the spring model with Coulomb's law to introduce repulsive forces between nodes, treating nodes as charged particles. These models optimize graph layouts in an iterative manner. More discussion can be found in Appendix B.

## 3 PRELIMINARY

### 3.1 BOLTZMANN DISTRIBUTION

The Boltzmann distribution (Boltzmann, 1868) plays a crucial role as a probability distribution function in the field of statistical mechanics. Its primary purpose is to quantify the probability of particles residing at specific energy levels. This distribution is commonly employed to describe how particles, such as molecules or atoms, are distributed among different states or energy levels in a system at thermal equilibrium. The mathematical representation of the Boltzmann distribution is as follows:

$$\mathbb{P}(E) = \frac{1}{Z} e^{-\frac{E}{kT}} \tag{3}$$

where $\mathbb{P}(E)$ is the probability of particles being in the energy level $E$, $Z$ is the normalization constant, known as the partition function, which ensures that the probability distribution sums to 1. $T$ is the temperature of the system in Kelvin, and $k$ is the temperature of the system in Kelvin, used to convert temperature $T$ from Kelvin to energy units.

## 3.2 EDGE-AWARE GNNS

Most current GNNs primarily emphasize message passing among node features. However, incorporating edge features into GNNs can enhance the topological layout of graphs to better suit downstream tasks. By modifying the element $\mathbf{C}_{v,u}^l$ located at position $(u, v)$ in the $l$-layer topological layout, an edge-aware GNN can be unified as:

$$\mathbf{C}_{v,u}^l = h_l \left( \mathbf{H}_{:v}^{(l)}, \mathbf{H}_{:u}^{(l)}, \mathbf{E}_{v,u}^{(l)}, \mathbf{A} \right) \tag{4}$$

where $\mathbf{H}_{:u}^{(l)}$ and $\mathbf{H}_{:v}^{(l)}$ denotes the node embedding of node $u$ and node $v$ in $l$-th layer, and $\mathbf{E}_{v,u}^{(l)}$ represents the edge embedding of the edge connecting nodes $u$ and $v$ in the $l$-th layer. $h_l$ is any trainable model parametrized by $l$.

## 4 METHODOLOGY

In this section, we detail the proposed method. An overview of the proposed method as well as a brief description is in Fig. 1. In general, DELs can be viewed as initial and supplementary edge features to a standard edge-aware GNN, conditioned on $\mathbf{A}$. In other words, DELs are *pre-calculated* only from connectivity $\mathbf{A}$. In Section 4.1, we introduce the way to sample a set of steady-state graph layouts following the Boltzmann distribution with local energy minima using a physics-driven graph layout algorithm. In Section 4.2, we present a simple yet effective edge embedding module to learn high-quality edge embeddings. In Section 4.3 and 4.4, we integrate these learned edge embeddings with several established GNN backbones.

### 4.1 GRAPH LAYOUT/CONFIGURATION SAMPLING

A typical layout representation of $\mathbf{C}$ is to utilize coordinates of nodes $\mathbf{P} \in \mathbb{R}^{n \times d}$ in $d$-dimensional space. Then there exists a mapping $\mathbf{C} = \mathbf{C}(\mathbf{P})$. Assuming that configuration $\mathbf{P}$ conditioned on a topology/connectivity $\mathbf{A}$ is associated with a global free energy $E(\mathbf{P}|\mathbf{A})$, we denote $E(\mathbf{P}) = E(\mathbf{P}|\mathbf{A})$ for brevity. One can sample a plausible configuration from an equipped Boltzmann distribution $\mathbb{P}(\mathbf{P}) \propto \exp(-E(\mathbf{P})/\alpha)$ using Langevin dynamics. Langevin dynamics originates from thermodynamics and has been widely used in optimization (Welling & Teh, 2011) and generative models (Song & Ermon, 2019; Shi et al., 2021). Concretely, by injecting i.i.d. Gaussian noise to the score term (Song & Ermon, 2019), the following update guarantees to sample $\mathbf{P} = \mathbf{P}_t$ from $\mathbb{P}(\mathbf{P})$ when $t \to \infty$:

$$\mathbf{P}_t = \mathbf{P}_{t-1} + \frac{\alpha}{2} \nabla_{\mathbf{P}} \log \mathbb{P}(\mathbf{P}_{t-1}) + \sqrt{\alpha} \epsilon_t, \qquad \text{where } \epsilon_t \sim \mathcal{N}(0, I) \tag{5}$$

where $\alpha \to 0_+$. Note to sample from Boltzmann distribution using Langevin dynamics only requests the score term $\nabla_{\mathbf{P}} \log \mathbb{P}(\mathbf{P}_{t-1})$. Additionally, as $\nabla_{\mathbf{P}} \log \mathbb{P}(\mathbf{P}_{t-1}) = -\nabla_{\mathbf{P}} E(\mathbf{P})$ (as partition function $Z$ does not dependent on $\mathbf{P}$), Eq. 5 amounts to performing gradient descent of $E$ w.r.t. $\mathbf{P}$ with proper additional noise. In this sense, as long as gradient term $\nabla_{\mathbf{P}} E(\mathbf{C})$ is well defined, one can readily sample $\mathbf{P}$ from the energy surface. Specifically, when $t \to \infty$ (i.e., $\mathbf{P}_t$ converges), $\mathbf{P} = \mathbf{P}_t$ can be viewed as a *steady-state configuration* of the free energy $E(\mathbf{P})$.

Empirically, we found that an approximation of the gradient $\nabla_{\mathbf{P}} E(\mathbf{P})$, indicating "appropriate update direction", is sufficient to sample satisfying $\mathbf{P}$ for the downstream tasks in specific energy settings, while a series of energy functions are optimized alike Eq. 5. Additionally, through all experiments, optimization with and without the noise term $\sqrt{\alpha} \epsilon_t$ generally leads to very similar diversity and performance. As such, we turn off this term and adopt standard graph layout algorithms to further accelerate the pre-processing. More details can be found in Appendix F and G. In the implementation of DEL, we employ two popularly utilized energy-based layouts in graph visualization falling into this category to sample $\mathbf{P}$, by further taking into account the computational overhead. We hereby introduce these algorithms. More empirical details can be found in Section 5.

**Fruchterman-Reingold Layout** (Fruchterman & Reingold, 1991), known as the spring-electrical model, is established by mimicking physical particle interactions consisting of attractive and repulsive forces. In this algorithm, nodes are treated as charged particles, and connections as springs, with an energy function combining electrical and spring potential energy. Although there does not exist

---

**Algorithm 1:** 2-dimensions Fruchterman-Reingold layout algorithm

---

**Input:** Graph $\mathcal{G} = (V, E)$ with $n$ nodes and $m$ edges, Iterations $N$, Ideal spring length $l_{ideal}$,
Attractive force coefficient $k_{attr}$, Repulsive force coefficient $k_{rep}$, Step size $\delta$.

**Output:** Node positions matrix $\mathbf{P} \in \mathbb{R}^{n \times 2}$.

1 Initialize node positions matrix $\mathbf{P}$ randomly, $\mathbf{p}_i$ denotes the $i$-th node position;
2 **for** $i \leftarrow 1$ **to** $N$ **do**
3      **foreach** *node* $v \in V$ **do**
4          Initialize net force acting on $v$: $\mathbf{F}_v \leftarrow (0, 0)$;
5          **foreach** *edge* $(v, u) \in E$ **do**
6              Calculate attractive force $\mathbf{F}_{attr}$ between $v$ and $u$ using Hooke's law:
             $\mathbf{F}_{attr} \leftarrow k_{attr} \cdot (||\mathbf{p}_u - \mathbf{p}_v|| - l_{ideal}) \cdot \frac{\mathbf{p}_u - \mathbf{p}_v}{||\mathbf{p}_u - \mathbf{p}_v||}$;
7              Update $\mathbf{F}_v$ with $\mathbf{F}_{attr}$;
8          **foreach** *node* $u \in V$ **do**
9              Calculate repulsive force $\mathbf{F}_{rep}$ between $v$ and $u$ using Coulomb's law:
             $\mathbf{F}_{rep} \leftarrow \frac{k_{rep}}{||\mathbf{p}_u - \mathbf{p}_v||^2} \cdot \frac{\mathbf{p}_u - \mathbf{p}_v}{||\mathbf{p}_u - \mathbf{p}_v||}$;
10              Update $\mathbf{F}_v$ with $\mathbf{F}_{rep}$;
11      **foreach** *node* $v \in V$ **do**
12          Update node position using net force $\mathbf{F}_v$: $\mathbf{p}_v \leftarrow \mathbf{p}_v + \delta \cdot \mathbf{F}_v$;

---

an explicit energy function, the iterative update implicitly guides the layout moving towards lower free energy. Such a process is demonstrated in Fig. 3 of Appendix C.1 and Algorithm 1 presents a summary of the Fruchterman-Reingold layout process. We term our method under this setting **DEL-F**.

**Kamada-Kawai Layout** (Kamada et al., 1989), on the other hand, only considers the spring forces on the edges. In this setting, the Kamada-Kawai Layout Algorithm connects all nodes in the graph with springs. The ideal lengths $l_{i,j}$ of these springs are determined by the shortest path distances within the original topology. Calculating the energy of the Kamada-Kawai Layout is straightforward. Given the coordinates of $n$ particles denoted as $\mathbf{P} \in \mathbb{R}^{n \times 2}$ and the strength $k_{i,j}$ of the spring between node $i$ and $j$ in the 2-dimensions Euclidean plane, the position $\mathbf{p}_i$ of $i$-th node represented by $(x_i, y_i)$, the associated energy is:

$$E = \sum_{i=1}^{n-1} \sum_{j=i+1}^{n} \frac{1}{2} k_{i,j} \left( (x_i - x_j)^2 + (y_i - y_j)^2 + l_{i,j}^2 - 2l_{i,j}\sqrt{(x_i - x_j)^2 + (y_i - y_j)^2} \right). \quad (6)$$

The algorithm aims to minimize the energy function $E(x_1, x_2, \ldots, x_n, y_1, y_2, \ldots, y_n)$ through the solution of a system of $2n$ simultaneous non-linear equations (Kobourov, 2012). To find the minimum of $E$ with respect to $x_m$ and $y_m$, the Newton-Raphson method is employed. At each step, the particle $\mathbf{p}_m$ with the largest value of $\Delta_m$ is selected, as defined below:

$$\Delta_m = \sqrt{\left(\frac{\partial E}{\partial x_m}\right)^2 + \left(\frac{\partial E}{\partial y_m}\right)^2} \quad (7)$$

For higher dimensions $dim$, the processing method is the same. The position of one node is fixed at each step selected by $\Delta_m$, and the final layout algorithm outputs a position matrix $\mathbf{P} \in \mathbb{R}^{n \times dim}$. More details of the Layout Algorithm can be found in Appendix C. Our method utilizing this layout is named as **DEL-K**.

Having sampled a set of steady-state configurations $\{\mathbf{P}^{(i)}\}_{i=1,\ldots,k}$ conditioned on $\mathbf{A}$, we turn our attention to how to exploit these steady-state configurations to boost graph representation learning. These graph configurations necessarily provide us with a global sketch of the energy surface.

*Remark.* Note both the Fruchterman-Reingold layout and Kamada-Kawai layout are optimized over node coordinates/configurations $\mathbf{P}$, thus we need extra effort to obtain $\mathbf{C} = \mathbf{C}(\mathbf{P})$, which will be detailed in the following sections. There are many optional layout algorithms that can potentially be combined with GNNs, such as spectral layout algorithms (Koren, 2003; Imre et al., 2020),

hyperbolic layout algorithms (Munzner & Burchard, 1995; Munzner, 1998; Riegler et al., 2016), etc., which are left to our future work.

## 4.2 EDGE FEATURES CONSTRUCTION VIA STEADY-STATE CONFIGURATIONS.

As stated, directly encoding $\mathbf{P}^{(i)}$ into a graph-level feature is inappropriate, as translation and rotation do not change its inherent structure. In this section, we propose an alternative approach by transforming $\mathbf{P}^{(i)}$ into edge features based on the pairwise distance between nodes, in accordance with the length of the springs in the graph. Specifically, the edge feature of each edge will be obtained by concatenating the length of this edge in all sampled layouts. The benefit of this edge feature is that it approximates the length distribution at which the spring reaches a steady state in all possible layouts. For the $i$-th layout, we can obtain an edge length matrix $\mathbf{L}_i \in \mathbb{R}^{n \times n}$ with $m$ non-zero elements:

$$\mathbf{L}_i = \mathbf{A} \odot \left\| \mathbf{P}^{(i)} - \mathbf{P}^{(i)\top} \right\|_2 \tag{8}$$

Now we can concatenate the edge length vectors of $k$ layouts and obtain the edge embedding $\mathbf{E} \in \mathbb{R}^{n \times n \times q}$ with $q$ hidden dimensions through a linear layer with weight $\mathbf{W} \in \mathbb{R}^{k \times q}$ and bias $\mathbf{b} \in \mathbb{R}^q$:

$$\mathbf{E} = \mathbf{W} \cdot ([\mathbf{L}_1, \mathbf{L}_2, \ldots, \mathbf{L}_k]) + \mathbf{b} \tag{9}$$

After obtaining the edge embedding $\mathbf{E}$, we then explore how to integrate it with the general form of GNNs.

## 4.3 GNNS WITH EDGE FEATURE

This edge embedding $\mathbf{E}$ derived from the steady-state layouts is an effective representation for integrating global information into the local message-passing process. We integrate the steady-state layout $\{\mathbf{C}^{(i)}\}_{i=1,\ldots,k}$ into multiple existing GNN frameworks via edge features $\mathbf{E}$, including GAT (Veličković et al., 2018), Graph Transformer (Shi et al., 2020), and GPS (Rampášek et al., 2022). Using GAT as an example, we have employed a standard approach that concatenates node features and edge features and applied an attention mechanism to obtain node representations. We can rewrite the general form Eq. 4 of edge-aware GAT as:

$$\mathbf{C}_{v,u}^l = \frac{a_{v,u}}{\sum_{k \in \bar{\mathcal{N}}(v)} a_{v,k}} \tag{10a}$$

$$\text{with} \quad a_{v,u} = \exp\left(\mathbf{a}^\top \sigma \left( h_l \left[ \mathbf{H}_{:v}^{(l)} \, \middle\| \, \mathbf{H}_{:u}^{(l)} \, \middle\| \, \mathbf{E}_{v,u}^{(l)} \right] \right) \right) \tag{10b}$$

where $\mathbf{H}_{:v}^{(l)}$ and $\mathbf{H}_{:u}^{(l)}$ represent the embeddings of nodes $u$ and $v$, and $\mathbf{E}_{v,u}^{(l)} \in \mathbb{R}^d$ represents the embedding of the edge between them in $l$-th layer. $h_l$ is a trainable model parametrized by $l$, $\mathbf{a}$ is a parameterized weight vector, where $\alpha_{u,v}$ represents the attention score between nodes $u$ and $v$, $\sigma$ is a non-linear activation function. Eq. 8, 9 and 10 altogether can be viewed as an approximation of integral calculation in Eq. 2. In other words, the mapping $\mathbf{C}(\mathbf{P})$ is not explicitly calculated, the integral instead is holistically approximated. From this, we can use steady-state layout distribution to enhance GNNs. Variants of other GNN models that consider edge features can be found in Appendix E.

## 4.4 DEL AS A PRE-PRECESSING STEP

Performing sampling on each GNN layer $l$ with Boltzmann distribution is impractical, due to not only the complexity, but also the instability and distribution shift introduced by hierarchical sampling. In our implementation, DEL is calculated only at the 0-th layer as:

$$\mathbf{H}^{(1)} = \sigma \left( \int \mathbf{C} \mathbf{H}^{(0)} \mathbf{W}^{(0)} \mathrm{d}\mathbb{P}(\mathbf{C}) \right) \tag{11}$$

Then $\mathbf{H}^{(1)}$ is fed to the subsequent layers as a standard GNN. In this sense, DEL serves as a pre-processing step ahead of GNNs, by sampling a set of $\mathbf{P}^{(i)}$ and conducting Eq. 11 without changing any other parts.

## 5 EXPERIMENTS

In this section, we extensively examine five key aspects. In Section 5.1, we present our primary results, elucidating the implementation details and performance of both our proposed method and the baseline methods. In Section 5.2, we investigate the influence of different graph layout states on the performance of DEL-F. In Section 5.3, we explore the application of DEL in high-dimensional spaces and analyze its performance in this context. In Section 5.4, we demonstrate the impact of layout numbers on the performance of DEL. Lastly, we evaluate the computational complexity of our method and provide insights into its resource requirements for different datasets.

### 5.1 MAIN RESULTS

**Datasets.** We perform experiments of graph-level tasks on widely used six datasets from TU-Dataset (Morris et al., 2020). Specifically, we employ two social network datasets, including IMDB-BINARY, IMDB-MULTI; and four bioinformatics datasets, including MUTAG, PROTEINS, NCI1, and D&D. More details of these datasets be found in Appendix A.

**Baselines.** We use five graph representation learning methods as baseline models: GMT (Baek et al., 2021), SEP-G (Wu et al., 2022), GIN (Xu et al., 2019b), Weisfeiler-Lehman sub-tree kernel (WL) (Shervashidze et al., 2011), GraphMAE (Hou et al., 2022). We also compare DEL with two edge feature construction methods. **Random distances** construct initial edge features with random distances and then apply our proposed edge embedding module to update the edge features. **MPNN** directly learns edge embeddings through node embeddings.

**Implementation details.** We applied DEL on three popular GNN backbones, including GAT (Veličković et al., 2018), Graph Transformer (Shi et al., 2020), and GPS (Rampášek et al., 2022). For GPS, we utilized the RWSE variant, which was considered to be more suitable for molecular data, while another variant, LapPE, is more suitable for images in previous work (Müller et al., 2023; Rampášek et al., 2022). Due to the presence of an MPNN mechanism in GPS, we opted not to use the corresponding MPNN variant. We developed GPS (DEL-F) and GPS (DEL-K) models that only leverage DEL for edge features. Additionally, we introduced GPS (RWSE, DEL-F) and GPS (RWSE, DEL-K) that incorporate both DEL and RWSE. To ensure a fair comparison, we followed the 10-fold cross-validation setting from previous work to obtain the model's performance (Zhang et al., 2018; Bianchi et al., 2020; Baek et al., 2021). We then ran all methods five times with different seeds to obtain the mean and standard deviation, which are reported in Table 1. More implementation details are discussed in Appendix A.

**Main Results & Analysis.** In Table 1, we can find that DEL-F and DEL-K can consistently improve the performance of the three GNN backbones on graph classification tasks, and DEL-F can achieve the best results most of the time. It's worth noting that previous studies (Sato et al., 2021; Wang & Zhang, 2022) have suggested that random features can sometimes improve GNN performance. As a comparison, we also explored the impact of random edge length initialization, essentially creating a random layout, within the context of DEL. However, our experiments reveal that, within our framework, arbitrary edge layouts tend to produce a detrimental effect. Additionally, the mechanism of MPNN seems to provide limited assistance within our framework. However, since DEL relies on the inherent graph topology for sampling the layout distribution, its improvements on datasets with a single consistent topology are somewhat limited, e.g., the graphs in IMDB-BINARY and IMDB-MULTI, which are ego networks with highly dense connections. Nevertheless, even within these densely connected ego networks, DEL can still identify certain distinct patterns through its layout sampling approach. For instance, in Fig. 6 of Appendix D it is observed that layout distribution tends to be more even for graphs with better symmetry and denser connections, indicating more stable topologies. Therefore, DEL still proves valuable in identifying graphs with highly similar topological structures through layout sampling.

### 5.2 EFFECT OF GRAPH LAYOUT STEADY STATE ON DEL-F PERFORMANCE.

In this section, we begin by drawing average energy curves for all the graphs in the MUTAG and NCI1 datasets, considering different number of layout iteration steps. Next, we use performance boxplots to illustrate how GAT combined with DEL-F performs under varying iteration counts. See Fig. 2. Each boxplot represents results from five times runs with different random seeds. By

Table 1: Test performance in five graph classification datasets. Highlighted in each cell are the top first and second results. We can observe that applying DELs to three common GNN backbones leads to significant improvement, outperforming other baseline methods.

| Method | MUTAG | NCI1 | PROTEINS | D&D | IMDB-BINARY | IMDB-MULTI |
|---|---|---|---|---|---|---|
| WL (Shervashidze et al., 2011) | 82.05 ± 0.36 | 82.19 ± 0.18 | 74.68 ± 0.49 | 79.78 ± 0.36 | 73.40 ± 4.63 | 49.33 ± 4.75 |
| GMT (Baek et al., 2021) | 83.44 ± 1.33 | 76.35 ± 2.62 | 75.09 ± 0.59 | 78.72 ± 0.59 | 73.48 ± 0.76 | 50.66 ± 0.82 |
| SEP-G (Wu et al., 2022) | 85.56 ± 1.09 | 78.35 ± 0.33 | 76.42 ± 0.39 | 77.98 ± 0.57 | 74.12 ± 0.56 | 51.53 ± 0.65 |
| GIN (Xu et al., 2019b) | 92.31 ± 0.87 | 80.26 ± 0.32 | 75.87 ± 0.35 | 75.83 ± 0.65 | 76.41 ± 0.93 | 52.70 ± 0.76 |
| GraphMAE (Hou et al., 2022) | 88.19 ± 1.26 | 80.40 ± 0.30 | 75.30 ± 0.39 | 79.42 ± 0.42 | 75.52 ± 0.66 | 51.63 ± 0.52 |
| GAT (Veličković et al., 2018) | 85.69 ± 1.06 | 77.98 ± 0.24 | 76.89 ± 0.37 | 78.12 ± 0.80 | 75.83 ± 0.51 | 52.65 ± 0.27 |
| +Random distance | 79.16 ± 1.77 | 71.46 ± 0.76 | 76.41 ± 0.54 | 77.32 ± 0.07 | 76.08 ± 0.48 | 52.60 ± 0.58 |
| +MPNN | 84.30 ± 1.21 | 77.28 ± 0.27 | 77.18 ± 0.41 | 77.64 ± 0.78 | 75.85 ± 0.48 | 52.86 ± 0.29 |
| +DEL-K | 86.52 ± 0.42 | 77.22 ± 0.15 | 77.34 ± 0.35 | 78.78 ± 0.35 | 76.86 ± 0.58 | 52.86 ± 0.33 |
| +DEL-F | 89.86 ± 1.21 | 78.59 ± 0.28 | 78.09 ± 0.27 | 78.11 ± 0.43 | 77.70 ± 0.32 | 53.00 ± 0.33 |
| Graph Transformer (Shi et al., 2020) | 84.02 ± 1.00 | 78.16 ± 0.38 | 77.04 ± 0.24 | 78.24 ± 0.46 | 76.43 ± 0.31 | 52.95 ± 0.35 |
| +Random distance | 79.72 ± 0.82 | 69.71 ± 0.12 | 75.27 ± 0.13 | 76.97 ± 0.51 | 75.27 ± 0.65 | 52.31 ± 0.45 |
| +MPNN | 83.75 ± 1.20 | 78.62 ± 0.23 | 77.29 ± 0.68 | 78.26 ± 0.57 | 76.51 ± 0.39 | 52.81 ± 0.25 |
| +DEL-K | 87.77 ± 0.12 | 78.67 ± 0.32 | 78.62 ± 0.68 | 79.25 ± 0.11 | 77.65 ± 0.49 | 53.54 ± 0.27 |
| +DEL-F | 92.22 ± 0.96 | 79.98 ± 0.25 | 78.58 ± 0.57 | 79.14 ± 0.82 | 78.10 ± 0.18 | 53.20 ± 0.30 |
| GPS(LapPE) (Rampášek et al., 2022) | 87.12 ± 2.37 | 73.88 ± 0.23 | 74.12 ± 1.23 | 77.43 ± 0.26 | 73.43 ± 0.56 | 50.31 ± 1.06 |
| GPS(RWSE) | 91.94 ± 1.50 | 79.97 ± 0.75 | 74.57 ± 0.79 | 78.65 ± 0.52 | 76.45 ± 1.03 | 51.84 ± 0.32 |
| GPS(RWSE, Random distance) | 91.66 ± 1.74 | 78.98 ± 0.41 | 73.26 ± 0.73 | 77.64 ± 0.42 | 75.98 ± 0.67 | 48.28 ± 0.84 |
| GPS(DEL-K) | 91.25 ± 1.89 | 82.53 ± 0.41 | 78.42 ± 0.59 | 80.56 ± 0.34 | 77.19 ± 0.96 | 52.65 ± 0.27 |
| GPS(DEL-F) | 90.97 ± 0.91 | 82.67 ± 0.35 | 77.68 ± 0.99 | 80.38 ± 0.33 | 76.87 ± 1.10 | 52.95 ± 0.64 |
| GPS(RWSE, DEL-K) | 91.67 ± 0.88 | 84.22 ± 0.20 | 78.04 ± 0.54 | 81.21 ± 0.78 | 77.27 ± 0.93 | 52.90 ± 0.49 |
| GPS(RWSE, DEL-F) | 92.23 ± 0.56 | 84.38 ± 0.23 | 78.26 ± 0.58 | 81.20 ± 0.51 | 77.04 ± 0.51 | 52.58 ± 0.65 |

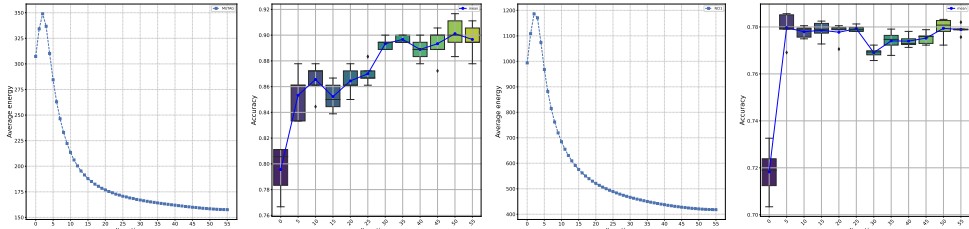

Figure 2: The average energy curve and graph classification performance of the MUTAG and NCI1 datasets under different layout iteration steps of DEL-F(GAT) are presented from left to right.

observing the average energy curves and performance boxplots of the two data sets, we can find that both datasets optimize the energy to be lower than the random layout in a very short period (less than five iterations) and continue to decline. As we continue to increase the number of layout iterations, we observe a consistent trend of decreasing layout energy and improving performance.

## 5.3 DEL IN HIGHER DIMENSIONAL SPACE

We then explore the impact of DELs in high-dimensional space. The two graph layout algorithms we employ are easily extendable to high-dimensional spaces by assigning initial random coordinates with dimensions higher than three. See results in Tab. 2. We found that DEL is not very sensitive to the dimension. However, DEL tends to achieve better results in two and three-dimensional spaces, possibly because it aligns better with the physical world. It's worth noting that there is no difference in the computational complexity of graph layout in different dimensions for the same number of iterations, and the actual time overhead is almost the same. However, higher dimensions might require more iterations to converge. Notice that another class of graph layout algorithms that easily extend to high-dimensional spaces are spectral graph layout algorithms, which we briefly discuss in Appendix C.

## 5.4 EFFECT OF DIFFERENT SAMPLING LAYOUT NUMBERS ON DEL PERFORMANCE

In this section, we focus on the impact of the number of sampled layouts on the performance of DEL on graph classification datasets. We chose three datasets for demonstration: MUTAG, PROTEINS, and NCI1, while using DEL-F in combination with three different GNN backbones. Table 3 presents part of the results, showing that as the number of sampled layouts increases, there is a noticeable

Table 2: The performance of DEL-F(GAT) with different layout dimensions. The tables about Graph transformer and GPS performance can be found in Appendix F.

| Dimensions | 2 | 3 | 4 | 5 | 6 |
|---|---|---|---|---|---|
| MUTAG | **89.86** $\pm$ **1.21** | 89.58 $\pm$ 0.91 | 89.30 $\pm$ 1.20 | 88.47 $\pm$ 1.20 | 88.75 $\pm$ 1.43 |
| NCI1 | 77.79 $\pm$ 0.42 | **78.59** $\pm$ **0.28** | 78.46 $\pm$ 0.31 | 78.35 $\pm$ 0.29 | 78.41 $\pm$ 0.20 |
| PROTEINS | **78.09** $\pm$ **0.27** | 77.29 $\pm$ 0.47 | 77.77 $\pm$ 0.43 | 78.06 $\pm$ 0.10 | 77.70 $\pm$ 0.36 |

Table 3: The performance of DEL-F (Graph Transformer) with different layout numbers. The tables about GAT and GPS performance can be found in Appendix F.

| N | 0 | 2 | 4 | 8 | 16 |
|---|---|---|---|---|---|
| MUTAG | 84.02 $\pm$ 1.00 | 89.86 $\pm$ 0.91 | 90.27 $\pm$ 0.92 | **92.22** $\pm$ **0.96** | 90.96 $\pm$ 1.68 |
| NCI1 | 78.16 $\pm$ 0.38 | 78.27 $\pm$ 0.62 | 79.00 $\pm$ 0.47 | 79.98 $\pm$ 0.25 | **80.12** $\pm$ **0.12** |
| PROTEIN | 77.04 $\pm$ 0.24 | 77.61 $\pm$ 0.46 | 77.88 $\pm$ 0.54 | 78.58 $\pm$ 0.57 | **78.78** $\pm$ **0.60** |

improvement in classification performance. This happens because DEL needs a sufficient number of sampled layouts to capture a wide range of energy distributions and better describe the inherent properties of the input topology. If we don't sample enough layouts, DEL can't comprehensively capture the energy surface, resulting in sub-optimal performance. But even with a small sample size, DEL still can perform better than the corresponding GNN backbone most of the time. Additional results for other GNN backbones can be found in Appendix F.

## 5.5 COMPUTATIONAL COMPLEXITY

For the preprocessing of DEL-F, each iteration of the basic algorithm computes $O(|E|)$ attractive forces and $O(|V|^2)$ repulsive forces. Therefore, the total computational complexity is $O(|E|+|V|^2)$. Johnson's algorithm (Johnson, 1977) is used to calculate the all-pair shortest path for DEL-K. The complexity of this calculation is $O(|V|^2 log|V| + |V| \cdot |E|)$. We have set the maximum iteration times to 50, following the settings of the previous work related to graph layout.

In addition to the theoretical computational complexity, we also provide the pre-processing time of DEL in practice when we sample eight layouts per graph, which can be found in Table 4. For datasets apart from D&D, we employ single-threaded processing, whereas, for D&D, we use eight parallel threads to reduce preprocessing time. It's worth noting that higher parallelism can further save preprocessing time.

Table 4: Runtime for DEL Preprocessing (seconds).

| Dataset | MUTAG | NCI1 | PROTEINS | DD | IMDB-BINARY | IMDB-MULTI |
|---|---|---|---|---|---|---|
| Avg. Degree | 39.58 | 64.60 | 145.63 | 1431.31 | 193.06 | 131.87 |
| Avg. Nodes | 17.9 | 29.8 | 39.1 | 284.3 | 19.8 | 13.0 |
| Graphs | 188 | 4,110 | 1,113 | 1,178 | 1,000 | 1,500 |
| DEL-F | 9.14 | 283.67 | 168.51 | 814.92 | 87.59 | 102.73 |
| DEL-K | 13.78 | 507.70 | 395.22 | 1944.53 | 154.72 | 124.11 |

## 6 CONCLUSIONS

We propose DEL, a novel method for graph representation learning. DELs are generated by sampling topological layouts from a Boltzmann distribution on an energy surface. This approach captures a wide spectrum of global graph information and improves the topological layout of graphs. In the implementation, DELs serve as a pre-processing step independent of subsequent GNN architectures, making them highly flexible and applicable. By integrating DELs into several GNNs, a significant improvement over a variety of datasets can be observed, which in turn supports its potential in practice.

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

## A    IMPLEMENTATION DETAIL

**Dataset.** The MUTAG dataset consists of seven kinds of graphs derived from 188 mutagenic compounds. NCI1 is a subset of balanced datasets from the National Cancer Institute, containing compounds screened for their ability to suppress human tumor cell growth. D&D contains protein structure graphs, with nodes representing amino acids and edges based on distance. PROTEINS dataset has nodes representing secondary structure elements, connected if neighboring in sequence or 3D space. IMDB-BINARY and IMDB-MULTI are movie collaboration datasets, where each graph represents actors/actresses and edges indicate their cooperation in a movie. Graphs are derived from specific movies, labeled with the movie genre. Except for the IMDB-BINARY and IMDB-MULTI datasets, other data sets have their own node features. In order to ensure a fair comparison, we use one-hot encodings of the degrees to construct the initial node features as in previous work for IMDB datasets (Baek et al., 2021; Wu et al., 2022).

**Model configuration.** For DEL application in three GNN backbones, we maintain the following hyperparameter settings: the learning rate is set to $5 \times 10^{-4}$, the node hidden size is set $\in \{32, 64\}$, the edge hidden size is set $\in \{16, 32\}$, the batch size is set $\in \{32, 64\}$, and weight decay is set to $1 \times 10^{-4}$. We use the early stopping criterion, where we stop the training if there is no further improvement in the validation loss during 25 epochs, Furthermore, the maximum number of epochs is set to 150. In DEL, each GNN backbone is stacked with two layers and employs the Relu non-linear activation function after each layer. The corresponding original GNN, random distance variant, and MPNN variant also maintain consistent settings.

## B    DISCUSSION ON GRAPH LAYOUT ALGORITHMS AND GNNS

The integration of graph layout and Graph Neural Networks (GNNs) is an area that has not been extensively explored, but there are potential connections between the two fields that can be mutually beneficial. For example, hyperbolic tree-based layout algorithms (Munzner & Burchard, 1995; Munzner, 1998; Riegler et al., 2016) aim to arrange hierarchical graphs in hyperbolic space, a concept that has also found exploration in the realm of GNNs (Liu et al., 2019; Zhang et al., 2021). Recent research in GNNs has highlighted the effectiveness of incorporating shortest path distance (SPD) to enhance the expressive power of GNNs. SP-GNN (Abboud et al., 2022), for instance, has demonstrated that GNNs using the k-SPD message passing mechanism outperform k-hop GNNs and the 1-WL test in terms of expressive power. Distance Encoding (Li et al., 2020) and SPD-WL (Zhang et al., 2022) have shown that incorporating SPD as node features can outperform the 1-WL test. Furthermore, SPD-WL has highlighted the ability to precisely identify cut edges within the graph by adding such features. Similarly, layout algorithms have recognized the importance of the shortest path distance early on. The Kamada-Kawai Layout Algorithm (Kamada et al., 1989) for example, connects all nodes in the graph with springs, and the ideal lengths $l_{i,j}$ of these springs are determined by the shortest path distances within the original topology. Another layout algorithm, the stress minimization (Gansner et al., 2005) algorithm also uses the shortest path distance as the ideal distance like the Kamada-Kawai Layout Algorithm, but it differs from the Kamada-Kawai Layout Algorithm in terms of the energy function it utilizes.

Overall, the integration of graph layout and GNNs presents a promising avenue for further exploration. The knowledge and techniques from both fields can complement each other.

## C    GRAPH LAYOUT ALGORITHM

In this section, we provide a detailed introduction to the graph layout algorithm.

**Spectral layout algorithm** (Koren, 2003; Imre et al., 2020) is a graph visualization algorithm that leverages spectral theory and eigenvector decomposition. It works by constructing the Laplacian matrix from the given graph, computing its eigenvectors, and using them as coordinates to position nodes in a multi-dimensional space. The spectral layout is known for preserving the global structure and relative distances in graphs, making it valuable for visualizing large networks and finding clustering patterns in various fields such as social network analysis, bioinformatics, and recommendation systems. It can also be customized and combined with other optimization methods to refine layout results based on specific requirements.

**The Kamada-Kawai algorithm** (Kamada et al., 1989) incorporates a spring-like model into its framework, which not only applies to connected nodes but also to unconnected ones. In this algorithm, nodes are treated as if they are connected by springs, with the ideal length of these springs set to the length of the shortest path between the nodes. This approach ensures that even unconnected nodes experience a form of spring-like force, contributing to the overall layout optimization. By considering these spring forces, the algorithm iteratively adjusts the positions of all nodes to minimize the total energy, resulting in a layout that not only shortens edge lengths for connected nodes but also positions unconnected nodes in an arrangement that reflects their shortest path distances. This holistic approach enhances the overall visualization and representation of the graph's structure. Algorithm 2 presents a summary of the Kamada-Kawai layout process.

---

**Algorithm 2:** Kamada-Kawai Algorithm

---

**Input:** Graph $\mathcal{G} = (V, E)$, Iterations $N$, Learning rate $\alpha$
**Output:** Node positions for a visually pleasing layout

1 Initialize node positions randomly and store them in $\mathbf{p}$;
2 Define constants for spring force $k_{attr}$;
3 Set a convergence threshold (e.g., small positive value);
4 Set a maximum number of iterations $N$;
5 Calculate the shortest path distances between all pairs of nodes and store them in $l_{\text{ideal}}$;
6 **for** $i \leftarrow 1$ **to** $N$ **do**
7     **foreach** *node* $v$ **do**
8         Initialize net force acting on $v$: $\mathbf{F}_v \leftarrow (0, 0)$;
9         **foreach** *edge* $e(u, v)$ **do**
10             Calculate spring force $\mathbf{F}_{attr}$ between $v$ and $u$ using Hooke's law:
$$\mathbf{F}_{attr} \leftarrow k_{attr} \cdot (||\mathbf{p}_u - \mathbf{p}_v|| - l_{\text{ideal}}[u][v]) \cdot \frac{\mathbf{p}_u - \mathbf{p}_v}{||\mathbf{p}_u - \mathbf{p}_v||};$$
11             Update $\mathbf{F}_v$ with $\mathbf{F}_{attr}$;
12     Calculate spring energy $\mathbf{E}_{\text{spring}}$ as the sum of spring forces:
$$E_{\text{spring}} = \frac{1}{2} \sum_{e(u,v) \in \mathbf{E}} k_{attr} \cdot (||\mathbf{p}_u - \mathbf{p}_v|| - l_{\text{ideal}}[u][v])^2$$
13     Calculate total energy $\mathbf{E} = \mathbf{E}_{\text{spring}}$;
14     **foreach** *node* $v \in V$ **do**
15         Compute the gradient of total energy $\mathbf{E}$ with respect to node positions: $\nabla \mathbf{E}_v \leftarrow \frac{\partial \mathbf{E}}{\partial \mathbf{p}_v}$;
16         Update node position using gradient descent: $\mathbf{p}_v \leftarrow \mathbf{p}_v - \alpha \cdot \nabla \mathbf{E}_v$;
17         Or use the Newton-Raphson method to select one node at a time and solve the equation to fix its position;
18     **if** $\mathbf{E} < convergence\_threshold$ **then**
19         **break**;

---

### C.1 LAYOUT ENERGY VISUALIZATION

Since the Fruchterman-Reingold layout algorithm does not explicitly optimize the layout energy, we separately visualized the average energy of all graphs in the six data sets under different iterations of the Fruchterman-Reingold layout algorithm in Fig. 3. The average energy consists of elastic potential energy and Coulomb potential energy. We found that the trend of energy changes is consistent. Starting from the random layout iteration, the layout energy first increases and then decreases, and finally converges to the local minimum energy layout. However, we can observe that the local optimal energy of the IMDB data set is higher than the random layout. This is because the graph in the IMDB data set is a highly densely connected ego network, and there are even some fully connected graphs. These densely connected graphs tend to form layouts that cover the entire plane, resulting in stability but also high potential energy.

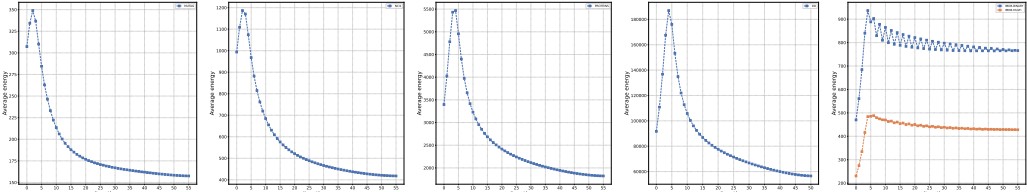

Figure 3: The layout energy change trend of the iterative process of the Fruchterman-Reingold layout algorithm for different datasets. From left to right present the average layout energy change trend of MUTAG, NCI1, PROTEINS, D&D, and IMDB datasets. In the IMDB dataset, all graphs consist of densely connected structures that tend to form layouts covering the entire plane, resulting in stability but also high potential energy.

## D  GRAPH LAYOUT VISUALIZATION

For a given graph, we calculate multiple layouts to explore different representations. The distance between layouts generated by the same graph is measured as the average Euclidean distance of all edge lengths within each layout. This information allows us to derive a layout distance matrix, denoted as $D \in \mathbb{R}^{n \times n}$, where $n$ is the number of layouts generated. To visualize this distance matrix, we employ a heatmap representation. To effectively compare the layout distance distributions across different graphs, we normalize the layout distances. This normalization process allows us to bring all the layout distances within a consistent range. It's worth noting that due to normalization and the original minimum layout distance being 0 (The diagonal elements of the original distance matrix.), all elements on the heatmap that are close to 0 or close to 1 signify a highly uniform layout distribution. This implies that all layouts are either very similar or that the layout distances are distributed evenly across the range, which is characteristic of highly concentrated or circular distributions. In order to gain a deeper insight into the layout distance distribution of a specific graph, we employ multidimensional scaling (MDS) techniques. MDS maps the layout distance matrix onto a low-dimensional space, such as a two-dimensional plane, while preserving the pairwise distances between layouts as much as possible. By visualizing the resulting MDS plot, we can observe the overall structure of the layout distance distribution, which can provide valuable insights into the characteristics of the graph. We have observed that graphs with distinct patterns often exhibit significantly different layout distance distributions. For example, in Fig. 4, our results suggest that graphs with a wider distribution of layout distances typically display folded layouts and long chains. On the other hand, graphs with a more uniform distribution of layout distances tend to feature a higher percentage of stable structures, such as rings. Similarly, in Fig. 5, we can observe that simple and stable network topologies yield uniform heatmaps and concentrated MDS clustering. These findings serve as evidence for the effectiveness of our approach in capturing and revealing patterns within molecular graphs.

It is important to note that the structure of social network graphs differs significantly from molecular graphs. Social network graphs often represent ego networks, which lack a distinct topological structure. To understand how our approach applies to these unique graph types, we have visualized the layout distance distribution of the IMDB dataset. The results, as shown in Fig. 6, clearly demonstrate that ego networks tend to exhibit highly uniform layouts due to their dense and symmetric connections. Despite the lack of a distinct structure in ego networks, we find that our topological layout algorithms can still benefit from these unique characteristics. In fact, we observe that layouts of ego networks tend to be more stable for graphs with better symmetry and denser connections, indicating more stable topologies. Furthermore, the use of MDS in two-dimensional space aids in distinguishing ego networks from other topologies and differentiating between various ego networks. These insights highlight the practical applications and advantages of our approach in different contexts.

In addition to the analysis of layout distance distributions, we also provide layout visualization examples of the six datasets we used in Fig. 7. These visualizations enable a better understanding of the unique topological structures present in different datasets. At the same time, we regularize the length of the edges in the layout and then color them. The longer the edges, the lighter the color. We can observe that the edge length is helpful in identifying those edges that connect different clusters.

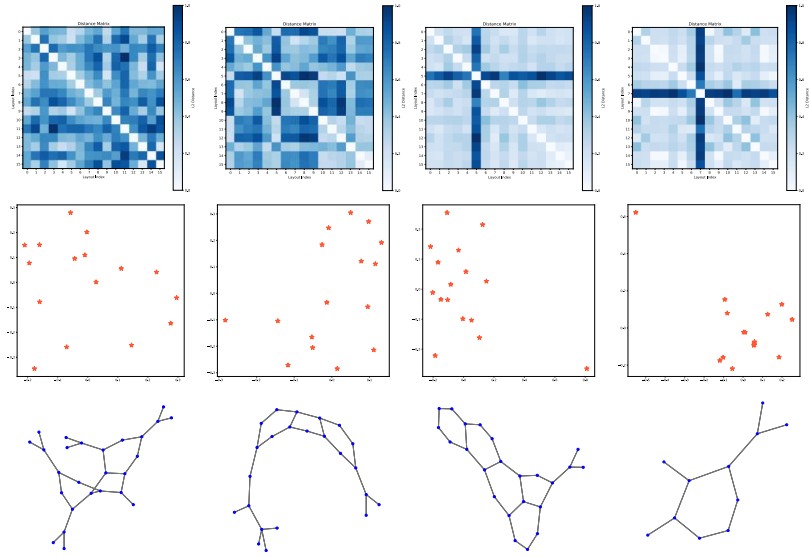

Figure 4: Heatmap, MDS layout and graph layout examples of MUTAG dataset.

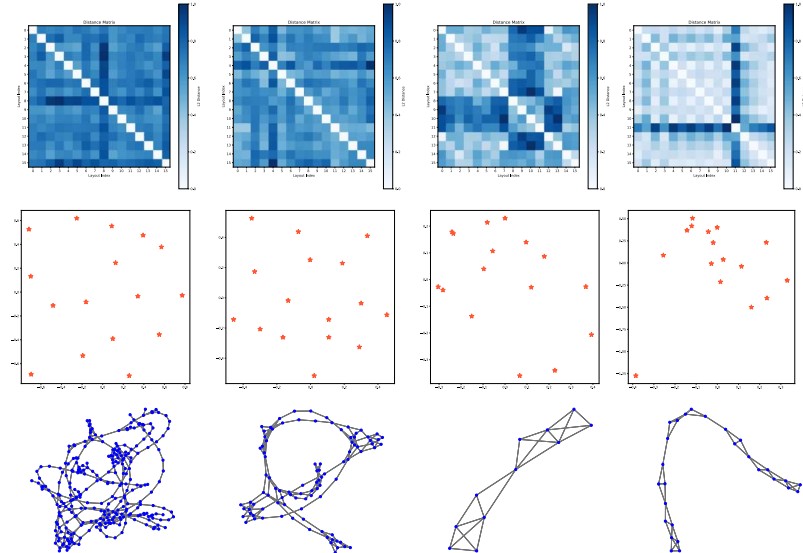

Figure 5: Heatmap, MDS layout and graph layout examples of PROTEINS dataset

# E    GENERIC GNNS FORMAT

Here, we provide the generic format of the graph transformer and GPS under DEL setting in Eq. 12 and Eq. 13:

$$\mathbf{C}_{v,u}^{(l)} = \mathrm{softmax}\left(\frac{\left(\mathbf{W}_1\mathbf{H}_{:v}^{(l)}\right)^{\top}\left(\mathbf{W}_2\mathbf{H}_{:u}^{(l)} + \mathbf{W}_3\mathbf{E}_{vu}^{(l)}\right)}{\sqrt{d}}\right) \tag{12}$$

Where $\mathbf{W}_1$, $\mathbf{W}_2$, and $\mathbf{W}_3$ are any trainable layers, $d$ denotes the hidden dimensions of node embedding.

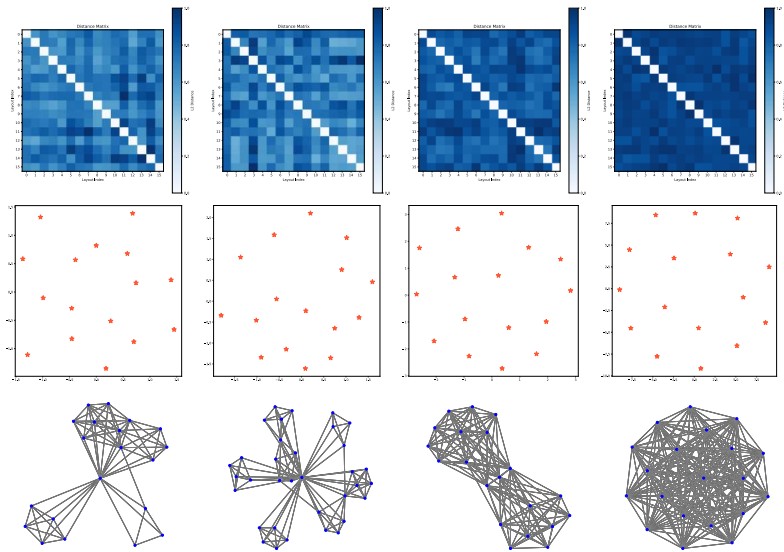

Figure 6: Heatmap, MDS layout and graph layout examples of IMDB-BINARY dataset

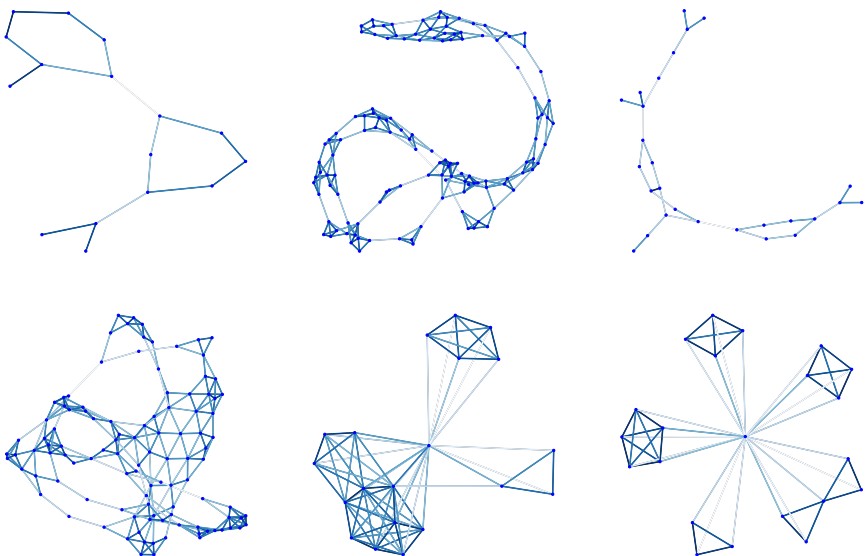

Figure 7: Graph layout examples for MUTAG, PROTEINS, NCI1, DD, IMDB-BINARY and IMDB-MULIT datasets.

$$\mathbf{C}_{v,u}^{(l)} = h_l \left( \mathbf{H}_{:v}^{(l)}, \mathbf{H}_{:u}^{(l)}, \mathbf{E}_{v,u}^{(l)}, \mathbf{A} \right) + \text{GlobalAttn}^l \left( \mathbf{H}_{:v}^{(l)} \right) \tag{13}$$

Where $\text{GlobalAttn}^l$ denotes the global attention layer in $l$-th layer which allows nodes to attend to all other nodes in a graph. $h_l$ is any trainable model parametrized by $l$.

## F   MORE RESULTS

In this section, we report four experimental results: First, The impact of DEL on graph classification tasks when graph transform and GPS are used as the backbones for graph layout in 2 to 6 dimensions in Table 5, 6. Second, The impact of using different sampling layout numbers on the graph classification task when DEL is employed with GAT or GPS as the backbones in Table 7 and 8. Third, The impact of noise addition during the sampling layout process on DEL performance in Table 9.

Finally, we evaluated DEL's performance on the ogbg-molhiv molecular property prediction dataset (Hu et al., 2020) (refer to Table 10), consisting of 40k graphs. Each graph represents a molecule, with nodes corresponding to atoms and edges representing chemical bonds. We observed a consistent improvement in ROC-AUC for both the validation and test sets. We employed a simple

| Dimensions | 2 | 3 | 4 | 5 | 6 |
|---|---|---|---|---|---|
| MUTAG | $91.38 \pm 1.00$ | $91.52 \pm 1.06$ | $\mathbf{91.80 \pm 0.67}$ | $92.22 \pm 0.96$ | $92.22 \pm 1.03$ |
| NCI1 | $79.98 \pm 0.25$ | $80.58 \pm 0.32$ | $\mathbf{80.60 \pm 0.31}$ | $80.56 \pm 0.47$ | $80.45 \pm 0.27$ |
| PROTEINS | $\mathbf{78.58} \pm 0.57$ | $78.35 \pm 0.28$ | $78.19 \pm 0.27$ | $78.17 \pm 0.54$ | $78.36 \pm 0.40$ |

Table 5: The performance of DEL-F (Graph Transformer) with different layout dimensions.

| Dimensions | 2 | 3 | 4 | 5 | 6 |
|---|---|---|---|---|---|
| MUTAG | $\mathbf{92.23 \pm 0.56}$ | $91.94 \pm 0.83$ | $91.80 \pm 0.60$ | $92.08 \pm 0.72$ | $91.94 \pm 0.92$ |
| NCI1 | $84.24 \pm 0.27$ | $84.23 \pm 0.25$ | $\mathbf{84.38 \pm 0.23}$ | $84.17 \pm 0.34$ | $84.25 \pm 0.41$ |
| PROTEINS | $78.26 \pm 0.58$ | $\mathbf{78.53 \pm 0.84}$ | $78.15 \pm 0.44$ | $78.17 \pm 0.98$ | $78.13 \pm 0.98$ |

Table 6: The performance of DEL-F (GPS) with different layout dimensions.

| N | 0 | 2 | 4 | 8 | 16 |
|---|---|---|---|---|---|
| MUTAG | $85.69 \pm 1.06$ | $87.08 \pm 0.46$ | $87.91 \pm 0.82$ | $\mathbf{89.86 \pm 1.21}$ | $87.23 \pm 0.82$ |
| NCI1 | $77.98 \pm 0.24$ | $76.63 \pm 0.18$ | $77.07 \pm 0.24$ | $\mathbf{78.59 \pm 0.28}$ | $78.30 \pm 0.22$ |
| PROTEINS | $76.89 \pm 0.37$ | $77.16 \pm 0.25$ | $77.34 \pm 0.69$ | $\mathbf{78.09 \pm 0.27}$ | $77.72 \pm 0.55$ |

Table 7: The performance of DEL-F (GAT) with different layout numbers.

| N | 0 | 2 | 4 | 8 | 16 |
|---|---|---|---|---|---|
| MUTAG | $91.94 \pm 1.50$ | $91.11 \pm 0.55$ | $91.80 \pm 0.51$ | $92.23 \pm 0.56$ | $\mathbf{92.22 \pm 0.87}$ |
| NCI1 | $79.97 \pm 0.75$ | $83.69 \pm 0.33$ | $83.97 \pm 0.34$ | $\mathbf{84.38 \pm 0.23}$ | $84.12 \pm 0.38$ |
| PROTEIN | $74.57 \pm 0.79$ | $77.73 \pm 0.88$ | $78.37 \pm 0.19$ | $\mathbf{78.26 \pm 0.58}$ | $77.72 \pm 0.88$ |

Table 8: The performance of DEL-F (GPS) with different layout numbers.

| Datasets | MUTAG | NCI1 |
|---|---|---|
| GAT(DEL-F) | $89.86 \pm 1.21$ | $78.59 \pm 0.28$ |
| +Noise | $89.30 \pm 1.32$ | $77.71 \pm 0.49$ |
| Graph transformer(DEL-F) | $92.22 \pm 0.96$ | $79.98 \pm 0.25$ |
| +Noise | $91.11 \pm 1.03$ | $78.51 \pm 0.20$ |
| GPS(DEL-F) | $92.23 \pm 0.56$ | $84.24 \pm 0.27$ |
| +Noise | $92.08 \pm 0.91$ | $83.85 \pm 0.34$ |

Table 9: Addition Noise of DEL-F. we can observe that optimization with and without the noise term $\epsilon_t \sim \mathcal{N}(0, 1e-3)$ generally leads to similar performance.

| ogbg-molhiv | Vaild AUROC | Test AUROC |
|---|---|---|
| GAT | $77.67 \pm 2.78$ | $74.45 \pm 1.53$ |
| GAT (DEL-F) | $\mathbf{78.99 \pm 1.69}$ | $\mathbf{75.27 \pm 1.32}$ |
| Graph transformer | $78.20 \pm 1.18$ | $76.51 \pm 0.93$ |
| Graph transformer (DEL-F) | $\mathbf{80.12 \pm 1.39}$ | $\mathbf{76.86 \pm 0.95}$ |
| GPS | $80.08 \pm 1.39$ | $76.71 \pm 1.12$ |
| GPS (DEL-F) | $\mathbf{82.57 \pm 0.95}$ | $\mathbf{76.93 \pm 1.00}$ |

Table 10: Valid and Test performance in ogbg-molhiv. Shown is the mean±std of 10 runs without any tricks. Note that ogbg-molhiv provides high-quality edge (chemical bonding) features, and we simply concatenated the edge features generated by DEL to it without using a complicated scheme.

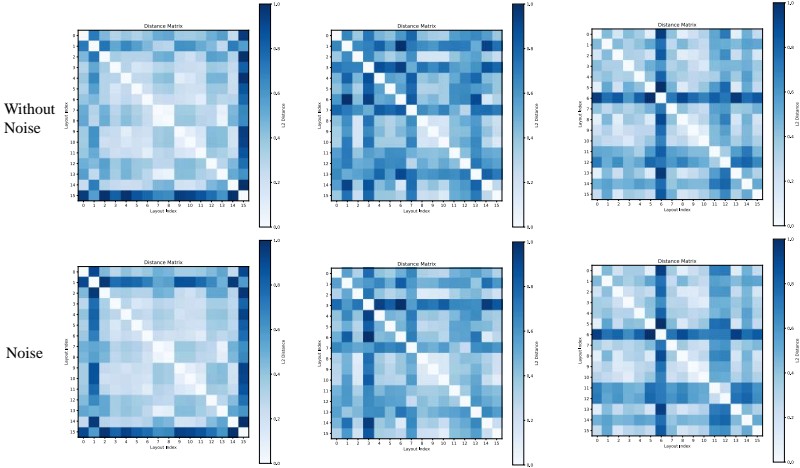

Figure 8: Heatmap of the distance between sampling layouts. The top of the figure shows the heatmap of the layout obtained by sampling with the standard layout algorithm, and the bottom shows the heatmap obtained by the layout algorithm with the added noise term $\epsilon_t \sim \mathcal{N}(0, 1e-3)$.

two-layer GNN architecture without special tricks, such as virtual nodes, etc., and defaulted to using Random-Walk Structural Encoding (RWSE) as a feature augmentation method.

## G ANALYSIS OF THE LAYOUT ALGORITHM WITH NOISE

In Section4.1 we used a standard graph layout algorithm instead of a graph layout algorithm with Gaussian noise using Langevin dynamics. In this section, we delve into a more detailed rationale for choosing this alternative approach.

A typical layout representation of $\mathbf{C}$ is to utilize coordinates of nodes $\mathbf{P} \in \mathbb{R}^{n \times d}$ in $d$-dimensional space. Then there exists a mapping $\mathbf{C} = \mathbf{C}(\mathbf{P})$. Assuming that configuration $\mathbf{P}$ conditioned on a topology/connectivity $\mathbf{A}$ is associated with a global free energy $E(\mathbf{P}|\mathbf{A})$, we denote $E(\mathbf{P}) = E(\mathbf{P}|\mathbf{A})$ for brevity. Our objective is to sample a plausible configuration from a Boltzmann distribution equipped with the Langevin dynamics. Specifically, we seek to sample from the distribution $\mathbb{P}(\mathbf{P}) \propto \exp(-E(\mathbf{P})/\alpha)$. In Section4.1 we have demonstrated that injecting i.i.d. Gaussian noise to the score term can theoretically achieve this goal.

However, in practical scenarios involving simple topologies like molecules and social networks from real-world applications, layout algorithms often produce layouts that are wide optima. Standard layout algorithms can be directly applied to achieve results similar to those obtained using Langevin dynamics, so we can use the standard layout algorithms directly as an approximation without the need for elaborate noise schemes. Figure 8 shows the heatmap of the distances between layouts, verifying that in real applications the addition of noise or not has a relatively small effect on the

distribution of the final sampled layouts. This reflects the characteristic of existing layout algorithms that tend to produce wide optima layouts. To reduce the computational burden caused by noise, we disable noise injection in our implementation.Table 9 further illustrates the limited impact of a basic noise addition strategy (using fixed Gaussian noise) on the downstream task. While a more sophisticated noise addition strategy could better assist the model in capturing the correct energy distribution, this is beyond the current scope and is left for future work.

