# OpenReview forum: "Graph Learning with Distributional Edge Layouts"
_ICLR.cc/2024/Conference — Submitted to ICLR 2024_

### Official Review · Reviewer_VZpe · 2023-10-26

**Soundness:** 3 good
**Presentation:** 3 good
**Contribution:** 2 fair
**Rating:** 5
**Confidence:** 3

**Summary:**

The paper proposes a method to sample layouts based on adjacency matrices. The layout can better capture the graph structures by injecting the sampled layout information into the GNN message-passing process. Several layout sampling methods are proposed. The authors evaluate the proposed methods on several benchmarks and increase the performance of the backbone graph learning models.  The layout computation process can be seen as a preprocessing step to boost the GNN's ability to learn graph structures.

**Strengths:**

The paper is well-written and easy to follow. The proposed approach to sample layout based on free energy seems well-attuned to real-world molecular applications. The proposed method naturally generates layout for each input graph which makes it easier to examine important substructures within the graph, such as functional groups in molecules.

**Weaknesses:**

- The evaluation is limited to smaller datasets (max 4k nodes). However, larger datasets such as QM9 and ZINC contain over 100k graphs. Showing performance increases on these datasets can further validate the method.

- I noticed that the computational efficiency is sub-optimal, requiring ~0.1 to 2 seconds per graph. Note that while preprocessing can save training time, it does not save inference time, which is more critical practically.

- The method is a preprocessing method that computes information unobtainable by MPNNs. While the experiments compare the method with some existing preprocessing methods, a clear comparison/analysis would help understand the mechanism behind the proposed method and why it is superior to RWPE. (In fact, it appears odd to me that the authors omit LapPE, since LapPE also considers global graph information (spectrum). Ideally, a comparison to LapPE should also be provided).

**Questions:**

Please refer to the weaknesses section.

---

> ### Author Response · Authors · 2023-11-20
>
> Thank you for your efforts in proposing these comments and questions.
> # Response to the weaknesses
> * In response to your concern regarding the evaluation's limitations, we appreciate your feedback and have made enhancements. The evaluation now incorporates results from the large-scale ogbg-molhiv dataset [1], presented in Table 10 on Page 20. We observe consistent performance enhancement on this dataset, aligning with existing results. This additional evaluation would provide more comprehensive and accurate support.
> | ogbg-molhiv               | Valid AUROC           | Test AUROC            |
> |---------------------------|-----------------------|-----------------------|
> | GAT                       | 77.67 $\pm$ 2.78      | 74.45 $\pm$ 1.53      |
> | GAT (DEL-F)                | **78.99 $\pm$ 1.69** | **75.27 $\pm$ 1.32** |
> | Graph transformer         | 78.20 $\pm$ 1.18      | 76.51 $\pm$ 0.93      |
> | Graph transformer (DEL-F)  | **80.12 $\pm$ 1.39** | **76.86 $\pm$ 0.95** |
> | GPS                       | 80.08 $\pm$ 1.39      | 76.71 $\pm$ 1.12      |
> | GPS (DEL-F)                | **82.57 $\pm$ 0.95** | **76.93 $\pm$ 1.00** |
>
> * In real-world applications, computing the graph layout for real-time test data does incur additional overhead. However, this overhead can often be mitigated through distributed computing. It's noteworthy that we report a minimal amount of preprocessing time using CPU threads (one or eight). For datasets other than D\&D, our processing speed for a single graph is typically around 0.1 seconds, while for D\&D, it is approximately 1 second. Given that the average number of nodes in existing graph classification datasets generally does not exceed that of D\&D, our method remains feasible in the majority of scenarios.
>
> * We did not deliberately disregard LapPE; our decision to focus on RWSE in the comparative experiment was guided by the conclusion of the paper [2]: "The benefits of the different encodings are very dataset dependant, with the random-walk structural encoding (RWSE) being more beneficial for molecular data and the Laplacian eigenvectors encodings (LapPE) being more beneficial in image superpixels." Therefore, we opted for the more advantageous RWSE for comparison. We will provide updated results for LapPE in Table 1.
>
> [1] Hu, Weihua, et al. "Open graph benchmark: Datasets for machine learning on graphs." Advances in neural information processing systems 33 (2020): 22118-22133.
>
> [2] Rampášek, Ladislav, et al. "Recipe for a general, powerful, scalable graph transformer." Advances in Neural Information Processing Systems 35 (2022): 14501-14515.

---

> ### Comment · Reviewer_VZpe · 2023-11-22
>
> After reading the author's response, some of my concerns (computation efficiency) are addressed. However, the proposed method minimally improves the performance of existing models on a larger dataset, ogbg-molhiv. The proposed method still appears to me as a variant of the preprocessing method, and the paper falls short of explaining why the technique outperforms the existing approach (RWSE), which limits its contribution. Hence, I am keeping my original rating.

---

### Official Review · Reviewer_8jRK · 2023-11-04

**Soundness:** 4 excellent
**Presentation:** 4 excellent
**Contribution:** 4 excellent
**Rating:** 6
**Confidence:** 2

**Summary:**

The paper introduces Distributional Edge Layouts (DEL) as a novel approach to graph representation learning where topological layouts are generated by sampling from a Boltzmann distribution. DEL is a  pre-processing technique which is independent of subsequent GNN architectures, demonstrating high flexibility and applicability. The integration of DELs into various GNNs yields improvements on diverse datasets, highlighting the practical potential of this approach. While the idea is innovative and shows promise, there are areas that need to be addressed to strengthen the paper

**Strengths:**

1) The concept of DEL, which involves sampling topological layouts from a Boltzmann distribution to improve graph representation learning, is a novel and interesting approach.

2) The paper presents a set of experiments using DEL in combination with various GNN backbones. The results demonstrate that DEL can enhance the performance of GNNs on diverse datasets, which is a notable strength.

3)  The paper provides clear explanations of the methodology, implementation details, and experimental setup.

**Weaknesses:**

- Lack of Theoretical Analysis: The paper could benefit from a more in-depth theoretical analysis of DEL. While the empirical results are promising, a theoretical framework that explains why DEL works and under what conditions would provide stronger support for the proposed approach.

- Dimensionality Exploration: The exploration of DEL in high-dimensional spaces is a valuable experiment. However, the paper would benefit from a more in-depth discussion of the implications of these findings.

- Sampling Layout Numbers: While the study addresses the impact of the number of sampled layouts on the performance, the explanations and discussions in this section are somewhat brief. More detailed insights into how the number of layouts affects the performance and potential trade-offs would enhance the paper's depth.

- Computational Complexity: The paper discusses the computational complexity of DEL, but there is room for a more comprehensive discussion, particularly concerning scalability. Discussing the implications of computational complexity in real-world, large-scale applications is crucial.

**Questions:**

why DEL works and under what conditions would provide stronger support for the proposed approach ?

Why does DEL perform better in lower dimensions, and what does this mean for practical applications?

What are the core theoretical principles behind DEL's effectiveness in graph representation learning?

Could you explain the trade-offs in choosing the number of sampled layouts for DEL, and are there practical constraints to consider?

---

> ### Author Response · Authors · 2023-11-20
>
> Thank you for your support and constructive comments.
> # Response to the weaknesses
> * Theoretical Analysis: Layout algorithms are seldom explored in graph representation learning, and proposing a comprehensive theoretical framework falls somewhat beyond the scope of this article. Nevertheless, DEL holds the potential to connect with certain theoretical frameworks. For instance, in Figure 7 on page 18, it is observed that DEL readily recognizes edges connecting different connected components. This observation could potentially be linked to spectral theory, as such edges often exert a significant influence on spectral distribution [1]. We will investigate more on the theoretical aspect in our future work.
>
> * Sampling Layout Numbers: The determination of the number of sampling layouts is contingent upon the characteristics of specific downstream tasks and datasets. In practical terms, ensuring an adequate number of sampling layouts is essential for the model to capture the potential energy distribution of the graph accurately. However, an excessively high number of layout samples can result in information redundancy and low efficiency, thereby impacting the performance of its practical use.
>
> * Computational Complexity: In real-world applications, graph sizes for graph classification tasks typically do not exceed those of the D\&D dataset. If the graph size is stable, the preprocessing overhead of DEL is linearly correlated with the number of graphs. The reported preprocessing times in the paper are based on the utilization of either 1 or 8 CPU threads; employing additional threads would further reduce preprocessing time.  Furthermore, for more efficient handling of node classification tasks of large-scale graphs, we can also combine DEL with some more efficient large-scale graph layout algorithms which capable of accommodating millions of nodes [3].
>
> * Regarding dimensionality exploration, Please refer to the 2nd response to the questions part.
>
> # Response to the questions
> Reviewer 8jRK’s question can be summarized as how DEL theoretically enhances graph representation learning and how to select important hyperparameters of DEL such as the number of samples and the dimension of the graph layout space.
>
> * DEL acquires global information, not attainable by MPNN through the sampling of the layouts based on potential energy distribution. This unique capability positions DEL as a potent enhancer of graph representation learning. In Appendix D, we present some initial analysis, reserving the construction of a more solid and comprehensive theoretical framework for future research.
>
> * Though there is not a ready theory interpreting why 2D and 3D layouts are sufficient to deliver good performance, we can regard graph spectral theory as an analogous counterpart. In graph spectral theory, the majority of Laplacian eigenvalues for a graph are very close to 0, while minor eigenvalues far from 0 are sufficient to depict the overall structure of the graph. This surprisingly aligns with the observation that 2D/3D layouts are adequate for capturing the main structure of the entire graph. Therefore, lacking specific prior knowledge, we prioritize performing DEL in a low-dimensional space. Simultaneously, DEL sampling multiple layouts in a low-dimensional space can further assist DEL in learning a more accurate energy distribution of the graph, leading to performance improvement. However, these are initial analyses, and more solid and comprehensive theoretical analyses will be left to future work.
>
> [1] Spielman, Daniel. "Spectral graph theory." Combinatorial scientific computing 18 (2012): 18.
>
> [2] Lin, Lu, Jinghui Chen, and Hongning Wang. "Spectral augmentation for self-supervised learning on graphs." ICLR 2023.
>
> [3] Zhu, Minfeng, et al. "DRGraph: An efficient graph layout algorithm for large-scale graphs by dimensionality reduction." IEEE Transactions on Visualization and Computer Graphics 27.2 (2020): 1666-1676.

---

### Official Review · Reviewer_YS2Q · 2023-11-06

**Soundness:** 2 fair
**Presentation:** 2 fair
**Contribution:** 1 poor
**Rating:** 3
**Confidence:** 5

**Summary:**

This paper introduces a new GNN architecture where the main idea is first to sample a graph layout and then use this layout for a standard message-passing GNN. The first step is done by sampling according to a Boltzmann distribution (this is not learned and done before the training of the GNN). It can be combined with any GNN architecture afterwards.

**Strengths:**

The idea of doing first a preprocessing on the graph in order to get a new layout is interesting as it can be coupled with any GNN architectures for downstream tasks.

**Weaknesses:**

The preprocessing step is not clearly defined in the paper. The authors use notions like Boltzmann distribution and Langevin dynamics, but if you look at the code, they only use a preprocessing using the Python library `networkx`. Algorithm 1 is unclear as it never uses the graph structure.

The evaluation is very weak because the datasets used (MUTAG, IMDB...) have been used extensively and are known to be very poor in order to compare the power of various architectures. Indeed, the differences between the various GNNs are very small.

**Questions:**

Please do not use complicated notions like Boltzmann distribution or Langevin dynamics unless these are really required. It is OK to use 'networkx` algorithms but then say it explicitly in the paper and remove the unnecessary parts. If I am making a mistake, please give me the lines of your code where your Algorithm 1 and Algorithm 2 are implemented.

For proper benchmarking of GNNs, you can use: Dwivedi, Vijay Prakash, et al. "Benchmarking graph neural networks." arXiv preprint arXiv:2003.00982 (2020). or more recent datasets.

---

> ### Author Response · Authors · 2023-11-20
>
> Thank you for your efforts in proposing these comments and questions.
> # Response to the weaknesses
> * We have provided related analysis and discussion about noise injection in our initial submission.
> We clearly define the preprocessing in Section 4.1 of the article, i.e., we use the standard graph layout algorithm as preprocessing, but we still provide the option to add noise in the code, e.g., you can find the Fruchterman Reingold algorithm with a noisy version in lines 44-61 and 248-308 in the utils.py implementation. In Section 4.1 we discussed that adding noise has little effect on the distribution of the sampled layouts as well as the downstream performance (shown in Table 9 of Appendix F at Page 19). Thus, to reduce the extra computational burden caused by noise, we turn off the noise injection in our implementation. As suggested by the reviewer YS2Q, we have reorganized our submission and discussed it in Appendix G to make this point clear.
> | ogbg-molhiv               | Valid AUROC           | Test AUROC            |
> |---------------------------|-----------------------|-----------------------|
> | GAT                       | 77.67 $\pm$ 2.78      | 74.45 $\pm$ 1.53      |
> | GAT (DEL-F)                | **78.99 $\pm$ 1.69** | **75.27 $\pm$ 1.32** |
> | Graph transformer         | 78.20 $\pm$ 1.18      | 76.51 $\pm$ 0.93      |
> | Graph transformer (DEL-F)  | **80.12 $\pm$ 1.39** | **76.86 $\pm$ 0.95** |
> | GPS                       | 80.08 $\pm$ 1.39      | 76.71 $\pm$ 1.12      |
> | GPS (DEL-F)                | **82.57 $\pm$ 0.95** | **76.93 $\pm$ 1.00** |
>
> * Algorithm 1 must consider the graph's topology, as the elastic force computed using Hooke's law applies only to nodes connected by edges. In response to your concerns, we have provided additional clarification in Algorithm 1, with the modifications highlighted in blue.
>
> * In response to your concern regarding the evaluation's limitations, we appreciate your feedback and have made enhancements. The evaluation now incorporates results from the large-scale ogbg-molhiv dataset [1], presented in Table 10 on Page 20. We observe consistent performance enhancement on this dataset, aligning with existing results. This additional evaluation would provide more comprehensive and accurate support.
>
> # Response to the questions
> * Please also refer to the 1st response to the weakness part.
> In practice, when dealing with simple topologies such as molecules and social networks from real-world applications, the layout algorithms tend to produce wide optima  [2], which are very close to the layouts sampled using noise-injected versions like Langevin dynamics (see Appendix G and Figure 8 in the manuscript).
> This allows for the use of standard layout algorithms as an efficient surrogate without the need for intricate noise schemes.
> In Figure 8 of Appendix G, we present new visualization results that confirm the close similarity of layout distributions obtained with or without noise sampling.
>
> [1] Hu, Weihua, et al. "Open graph benchmark: Datasets for machine learning on graphs." Advances in neural information processing systems 33 (2020): 22118-22133.
>
> [2] Fruchterman, Thomas MJ, and Edward M. Reingold. "Graph drawing by force‐directed placement." Software: Practice and experience 21.11 (1991): 1129-1164.

---

> > ### Comment · Reviewer_YS2Q · 2023-11-21
> > **Thank you for your response**
> >
> > It looks like my initial understanding of the paper is correct, and I still think that most of the discussion concerning layout should be simplified as, in the end, simple `networkx` routines are used. Written as is, the paper is very misleading, and its contribution is weak.

---

> > > ### Author Response · Authors · 2023-11-21
> > >
> > > 1. If reviewer YS2Q read our submission carefully, it would be readily identified that our claim is "...we propose to cope with layouts by introducing an associated distribution over $\mathbf{C}$" (in Eq. 2). This is our claimed contribution and we believe this is the first work that equips layout in GNNs with a latent distribution $\mathbb{P}(\mathbf{C})$, where $\mathrm{networkx}$ happens to provide a convenient way to sample such layouts under various energy surfaces. One can incorporate other energy functions as well, which may appear in $\mathrm{networkx}$ or not. Therefore, we respectfully disagree with reviewer YS2Q to assert that our contribution mainly comes from $\mathrm{networkx}$. It is also appreciated that reviewers 8jRK and VZpe confirmed the novelty and the contribution of our work.
> > >
> > > 2. We turn off the noise because the performance, as well as the sampled distribution, is very similar to the noise-free version but with an extra computational burden. And we frankly reported this. Indeed, turning on the noise aligns with the Langevin Dynamics and also brings about performance improvement, but we believe only reporting noise-injected DEL is misleading. There is no reason to sacrifice computational time for a "better story", especially when some of the readers may really use DEL with the selected energy functions.

---

### Meta-Review · Area_Chair_kXiE · 2023-12-06

**Metareview:**

The paper describes a new GNN architecture called Distributional Edge Layouts (DEL). DEL involves sampling topological layouts for graphs from a Boltzmann distribution as a preprocessing step. These layouts are then used to enhance the performance of various GNN architectures on downstream tasks. The paper presents experiments demonstrating that DEL can improve the performance of GNNs on different datasets.

Strengths of the paper:
* DEL is presented as a preprocessing technique that can be integrated with various GNN architectures, making it flexible and applicable to a wide range of tasks and datasets.
* The paper is well-written and presents the methodology, implementation details, and experimental setup clearly, making it easy to follow.

Weaknesses of the paper:
* The paper lacks a deep theoretical analysis of DEL. A theoretical framework explaining why DEL works and under what conditions would enhance the paper's credibility.
* The paper's evaluation is limited to smaller datasets, and testing on larger datasets would provide further validation of the method's effectiveness.
* Reviewers have raised concerns about the clarity of the preprocessing step, with some suggesting that the paper should explicitly state the use of "networkx" algorithms and remove unnecessary complex notions.
* The paper's choice of benchmark datasets has been criticized for not being the most suitable for evaluating GNN performance. Reviewers suggest using more recent and comprehensive benchmark datasets for better benchmarking.

**Justification For Why Not Higher Score:**

The paper introduces an innovative approach, DEL, for enhancing GNN performance through preprocessing. While the concept is promising, the paper could benefit from a stronger theoretical foundation, a more in-depth exploration of key factors, and a clearer presentation of the preprocessing step. Additionally, testing on larger and more relevant datasets and addressing computational efficiency concerns would strengthen the paper's contributions.

**Justification For Why Not Lower Score:**

N/A

---

### Decision · Program_Chairs · 2024-01-16

Reject